# Top Rank Optimization in Linear Time

**Nan Li**[1]     **Rong Jin**[2]     **Zhi-Hua Zhou**[1]
[1]National Key Laboratory for Novel Software Technology,
Nanjing University, Nanjing 210023, China
[2]Department of Computer Science and Engineering,
Michigan State University, East Lansing, MI 48824
`{lin,zhouzh}@lamda.nju.edu.cn` `rongjin@cse.msu.edu`

## Abstract

Bipartite ranking aims to learn a real-valued ranking function that orders positive instances before negative instances. Recent efforts of bipartite ranking are focused on optimizing ranking accuracy at the top of the ranked list. Most existing approaches are either to optimize task specific metrics or to extend the rank loss by emphasizing more on the error associated with the top ranked instances, leading to a high computational cost that is super-linear in the number of training instances. We propose a highly efficient approach, titled **TopPush**, for optimizing accuracy at the top that has computational complexity linear in the number of training instances. We present a novel analysis that bounds the generalization error for the top ranked instances for the proposed approach. Empirical study shows that the proposed approach is highly competitive to the state-of-the-art approaches and is 10-100 times faster.

## 1 Introduction

Bipartite ranking aims to learn a real-valued ranking function that places positive instances above negative instances. It has attracted much attention because of its applications in several areas such as information retrieval and recommender systems [32, 25]. Many ranking methods have been developed for bipartite ranking, and most of them are essentially based on pairwise ranking. These algorithms reduce the ranking problem into a binary classification problem by treating each positive-negative instance pair as a single object to be classified [16, 12, 5, 39, 38, 33, 1, 3]. Since the number of instance pairs can grow quadratically in the number of training instances, one limitation of these methods is their high computational costs, making them not scalable to large datasets.

Considering that for applications such as document retrieval and recommender systems, only the top ranked instances will be examined by users, there has been a growing interest in learning ranking functions that perform especially well at the top of the ranked list [7, 39, 38, 33, 1, 3, 27, 40]. Most of these approaches can be categorized into two groups. The first group maximizes the ranking accuracy at the top of the ranked list by optimizing task specific metrics [17, 21, 23, 40], such as average precision (AP) [42], NDCG [39] and partial AUC [27, 28]. The main limitation of these methods is that they often result in non-convex optimization problems that are difficult to solve efficiently. Structural SVM [37] addresses this issue by translating the non-convexity into an exponential number of constraints. It can still be computationally challenging because it usually requires to search for the most violated constraint at each iteration of optimization. In addition, these methods are statistically inconsistent [36, 21], leading to suboptimal solutions. The second group of methods are based on pairwise ranking. They design special convex loss functions that place more penalties on the ranking errors related to the top ranked instances [38, 33, 1]. Since these methods are based on pairwise ranking, their computational costs are usually proportional to the number of positive-negative instance pairs, making them unattractive for large datasets.

In this paper, we address the computational challenge of bipartite ranking by designing a ranking algorithm, named **TopPush**, that can efficiently optimize the ranking accuracy at the top. The key feature of the proposed TopPush algorithm is that its time complexity is only *linear* in the number of training instances. This is in contrast to most existing methods for bipartite ranking whose computational costs depend on the number of instance pairs. Moreover, we develop novel analysis for bipartite ranking. One deficiency of the existing theoretical studies [33, 1] on bipartite ranking is that they try to bound the probability for a positive instance to be ranked before *any* negative instance, leading to relatively pessimistic bounds. We overcome this limitation by bounding the probability of ranking a positive instance before *most* negative instances, and show that TopPush is effective in placing positive instances at the top of a ranked list. Extensive empirical study shows that TopPush is computationally more efficient than most ranking algorithms, and yields comparable performance as the state-of-the-art approaches that maximize the ranking accuracy at the top.

The rest of this paper is organized as follows. Section 2 introduces the preliminaries of bipartite ranking, and addresses the difference between AUC optimization and maximizing accuracy at the top. Section 3 presents the proposed TopPush algorithm and its key theoretical properties. Section 4 summarizes the empirical study, and Section 5 concludes this work with future directions.

## 2  Bipartite Ranking: AUC vs. Accuracy at the Top

Let $\mathcal{X} = \{\mathbf{x} \in \mathbb{R}^d : \|\mathbf{x}\| \leq 1\}$ be the instance space. Let $S = S_+ \cup S_-$ be a set of training instances, where $S_+ = \{\mathbf{x}_i^+ \in \mathcal{X}\}_{i=1}^m$ and $S_- = \{\mathbf{x}_i^- \in \mathcal{X}\}_{i=1}^n$ include $m$ positive instances and $n$ negative instances independently sampled from distributions $\mathcal{P}_+$ and $\mathcal{P}_-$, respectively. The goal of bipartite ranking is to learn a ranking function $f : \mathcal{X} \mapsto \mathbb{R}$ that is likely to place a positive instance before most negative ones. In the literature, bipartite ranking has found applications in many domains [32, 25], and its theoretical properties have been examined by several studies [2, 6, 20, 26].

AUC is a commonly used evaluation metric for bipartite ranking [15, 9]. By exploring its equivalence to Wilcoxon-Mann-Whitney statistic [15], many ranking algorithms have been developed to optimize AUC by minimizing the ranking loss defined as

$$\mathcal{L}_{\text{rank}}(f; S) = \frac{1}{mn} \sum\nolimits_{i=1}^m \sum\nolimits_{j=1}^n \mathbb{I}\big(f(\mathbf{x}_i^+) \leq f(\mathbf{x}_j^-)\big) , \qquad (1)$$

where $\mathbb{I}(\cdot)$ is the indicator function. Other than a few special loss functions (e.g., exponential and logistic loss) [33, 20], most of these methods need to enumerate all the positive-negative instance pairs, making them unattractive for large datasets. Various methods have been developed to address this computational challenge [43, 13].

Recently, there is a growing interest on optimizing ranking accuracy at the top [7, 3]. Maximizing AUC is not suitable for this goal as indicated by the analysis in [7]. To address this challenge, we propose to maximize the number of positive instances that are ranked before the first negative instance, which is known as *positives at the top* [33, 1, 3]. We can translate this objective into the minimization of the following loss

$$\mathcal{L}(f; S) = \frac{1}{m} \sum\nolimits_{i=1}^m \mathbb{I}\Big(f(\mathbf{x}_i^+) \leq \max_{1 \leq j \leq n} f(\mathbf{x}_j^-)\Big) . \qquad (2)$$

which computes the fraction of positive instances ranked below the top-ranked negative instance. By minimizing the loss in (2), we essentially push negative instances away from the top of the ranked list, leading to more positive ones placed at the top. We note that (2) is fundamentally different from AUC optimization as AUC does not focus on the ranking accuracy at the top. More discussion about the relationship between (1) and (2) can be found in the longer version of the paper [22].

To design practical learning algorithms, we replace the indicator function in (2) with its convex surrogate, leading to the following loss function

$$\mathcal{L}^\ell(f; S) = \frac{1}{m} \sum\nolimits_{i=1}^m \ell\Big(\max_{1 \leq j \leq n} f(\mathbf{x}_j^-) - f(\mathbf{x}_i^+)\Big) , \qquad (3)$$

where $\ell(\cdot)$ is a convex loss function that is non-decreasing[1] and differentiable. Examples of such loss functions include truncated quadratic loss $\ell(z) = [1 + z]_+^2$, exponential loss $\ell(z) = e^z$, or

logistic loss $\ell(z) = \log(1 + e^z)$. In the discussion below, we restrict ourselves to the truncated quadratic loss, though most of our analysis applies to others.

It is easy to verify that the loss $\mathcal{L}^\ell(f; S)$ in (3) is equivalent to the loss used in InfinitePush [1] (a special case of $P$-norm Push [33]), i.e.,

$$\mathcal{L}^\ell_\infty(f; S) = \max_{1 \leq j \leq n} \frac{1}{m} \sum\nolimits_{i=1}^m \ell\big(f(\mathbf{x}_j^-) - f(\mathbf{x}_i^+)\big) . \tag{4}$$

The apparent advantage of employing $\mathcal{L}^\ell(f; S)$ instead of $\mathcal{L}^\ell_\infty(f; S)$ is that it only needs to evaluate on $m$ positive-negative instance pairs, whereas the later needs to enumerate all the $mn$ instance pairs. As a result, the number of dual variables induced by $\mathcal{L}^\ell(f; S)$ is $n + m$, linear in the number of training instances, which is significantly smaller than $mn$, the number of dual variables induced by $\mathcal{L}^\ell_\infty(f; S)$ [1, 31]. It is this difference that makes the proposed algorithm achieve a computational complexity linear in the number of training instances and therefore be more efficiently than the existing algorithms for most state-of-the-art algorithms for bipartite ranking.

# 3   TopPush for Optimizing Top Accuracy

We first present a learning algorithm to minimize the loss function in (3), and then the computational complexity and performance guarantee for the proposed algorithm.

## 3.1   Dual Formulation

We consider linear ranking function[2], i.e., $f(\mathbf{x}) = \mathbf{w}^\top \mathbf{x}$, where $\mathbf{w} \in \mathbb{R}^d$ is the weight vector to be learned. As a result, the learning problem is given by the following optimization problem

$$\min_{\mathbf{w}} \quad \frac{\lambda}{2} \|\mathbf{w}\|^2 + \frac{1}{m} \sum\nolimits_{i=1}^m \ell\Big( \max_{1 \leq j \leq n} \mathbf{w}^\top \mathbf{x}_j^- - \mathbf{w}^\top \mathbf{x}_i^+ \Big) , \tag{5}$$

where $\lambda > 0$ is a regularization parameter. Directly minimizing the objective in (5) can be challenging because of the max operator in the loss function. We address this challenge by developing a dual formulation for (5). Specifically, given a convex and differentiable function $\ell(z)$, we can rewrite it in its convex conjugate form as $\ell(z) = \max_{\alpha \in \Omega} \alpha z - \ell_*(\alpha)$ , where $\ell_*(\alpha)$ is the convex conjugate of $\ell(z)$ and $\Omega$ is the domain of dual variable [4]. For example, the convex conjugate of truncated quadratic loss is $\ell_*(\alpha) = -\alpha + \alpha^2/4$ with $\Omega = \mathbb{R}_+$. We note that dual form has been widely used to improve computational efficiency [35] and connect different styles of learning algorithms [19]. Here we exploit it to overcome the difficulty caused by max operator. The dual form of (5) is given in the following theorem, whose detailed proof can be found in the longer version [22].

**Theorem 1.** *Define* $\mathbf{X}^+ = (\mathbf{x}_1^+, \ldots, \mathbf{x}_m^+)^\top$ *and* $\mathbf{X}^- = (\mathbf{x}_1^-, \ldots, \mathbf{x}_n^-)^\top$, *the dual problem of (5) is*

$$\min_{(\boldsymbol{\alpha}, \boldsymbol{\beta}) \in \Xi} \quad g(\boldsymbol{\alpha}, \boldsymbol{\beta}) = \frac{1}{2\lambda m} \|\boldsymbol{\alpha}^\top \mathbf{X}^+ - \boldsymbol{\beta}^\top \mathbf{X}^-\|^2 + \sum\nolimits_{i=1}^m \ell_*(\alpha_i) \tag{6}$$

*where* $\boldsymbol{\alpha}$ *and* $\boldsymbol{\beta}$ *are dual variables, and the domain* $\Xi$ *is defined as*

$$\Xi \quad = \quad \big\{ \boldsymbol{\alpha} \in \mathbb{R}_+^m, \, \boldsymbol{\beta} \in \mathbb{R}_+^n : \, \mathbf{1}_m^\top \boldsymbol{\alpha} = \mathbf{1}_n^\top \boldsymbol{\beta} \big\}.$$

*Let* $\boldsymbol{\alpha}^*$ *and* $\boldsymbol{\beta}^*$ *be the optimal solution to the dual problem (6). Then, the optimal solution* $\mathbf{w}^*$ *to the primal problem in (5) is given by*

$$\mathbf{w}^* = \frac{1}{\lambda m} \big( \boldsymbol{a}^{*\top} \mathbf{X}^+ - \boldsymbol{\beta}^{*\top} \mathbf{X}^- \big) . \tag{7}$$

**Remark** The key feature of the dual problem in (6) is that the number of dual variables is $m + n$, leading to a linear time ranking algorithm. This is in contrast to the InfinitPush algorithm in [1] that introduces $mn$ dual variables and a higher computational cost. In addition, the objective function in (6) is smooth if the convex conjugate $\ell_*(\cdot)$ is smooth, which is true for many common loss functions (e.g., truncated quadratic loss and logistic loss). It is well known in the literature of optimization [4] that an $O(1/T^2)$ convergence rate can be achieved if the objective function is smooth, where $T$ is the number of iterations; this also helps in designing efficient learning algorithm.

## 3.2 Linear Time Bipartite Ranking

According to Theorem 1, to learn a ranking function $f(\mathbf{w})$, it is sufficient to learn the dual variables $\boldsymbol{\alpha}$ and $\boldsymbol{\beta}$ by solving the problem in (6). For this purpose, we adopt the accelerated gradient method due to its light computation per iteration, and refer the obtained algorithm as **TopPush**. Specifically, we choose the Nesterov's method [30, 29] that achieves an optimal convergence rate $O(1/T^2)$ for smooth objective function. One of the key features of the Nesterov's method is that it maintains two sequences of solutions: $\{(\boldsymbol{\alpha}_k, \boldsymbol{\beta}_k)\}$ and $\{(\mathbf{s}_k^\alpha; \mathbf{s}_k^\beta)\}$, where the sequence of auxiliary solutions $\{(\mathbf{s}_k^\alpha; \mathbf{s}_k^\beta)\}$ is introduced to exploit the smoothness of the objective to achieve a faster convergence rate. Algorithm 1 shows the key steps[3] of the Nesterov's method for solving the problem in (6), where the gradients of the objective function $g(\boldsymbol{\alpha}, \boldsymbol{\beta})$ can be efficiently computed as

$$\nabla_{\boldsymbol{\alpha}} g(\boldsymbol{\alpha}, \boldsymbol{\beta}) = \mathbf{X}^+ \boldsymbol{\nu}^\top / \lambda m + \ell'_*(\boldsymbol{\alpha}) , \quad \nabla_{\boldsymbol{\beta}} g(\boldsymbol{\alpha}, \boldsymbol{\beta}) = -\mathbf{X}^- \boldsymbol{\nu}^\top / \lambda m . \tag{8}$$

where $\boldsymbol{\nu} = \boldsymbol{\alpha}^\top \mathbf{X}^+ - \boldsymbol{\beta}^\top \mathbf{X}^-$ and $\ell'_*(\cdot)$ is the derivative of $\ell_*(\cdot)$.

---

**Algorithm 1** The TopPush Algorithm

---

**Input:** $\mathbf{X}^+ \in \mathbb{R}^{m \times d}$, $\mathbf{X}^- \in \mathbb{R}^{n \times d}$, $\lambda$, $\epsilon$
**Output:** $\mathbf{w}$
1: initialize $\boldsymbol{\alpha}_1 = \boldsymbol{\alpha}_0 = \mathbf{0}_m$, $\boldsymbol{\beta}_1 = \boldsymbol{\beta}_0 = \mathbf{0}_n$, and let $t_{-1} = 0$, $t_0 = 1$, $L_0 = \frac{1}{m+n}$
2: **repeat** for $k = 1, 2, \ldots$
3:   compute $\mathbf{s}_k^a = \boldsymbol{\alpha}_k + \omega_k (\boldsymbol{\alpha}_k - \boldsymbol{\alpha}_{k-1})$ and $\mathbf{s}_k^\beta = \boldsymbol{\beta}_k + \omega_k (\boldsymbol{\beta}_k - \boldsymbol{\beta}_{k-1})$, where $\omega_k = \frac{t_{k-2}-1}{t_{k-1}}$
4:   compute $\mathbf{g}_{\boldsymbol{\alpha}} = \nabla_{\boldsymbol{\alpha}} g(\mathbf{s}_k^\alpha, \mathbf{s}_k^\beta)$ and $\mathbf{g}_{\boldsymbol{\beta}} = \nabla_{\boldsymbol{\beta}} g(\mathbf{s}_k^\alpha, \mathbf{s}_k^\beta)$ based on (8)
5:   find $L_k > L_{k-1}$ such that $g(\boldsymbol{\alpha}_{k+1}, \boldsymbol{\beta}_{k+1}) > g(\mathbf{s}_k^\alpha, \mathbf{s}_k^\beta) + (\|\mathbf{g}_{\boldsymbol{\alpha}}\|^2 + \|\mathbf{g}_{\boldsymbol{\beta}}\|^2)/(2L_k)$, where
       $[\boldsymbol{\alpha}_{k+1}; \boldsymbol{\beta}_{k+1}] = \pi_\Xi([\boldsymbol{\alpha}'_{k+1}; \boldsymbol{\beta}'_{k+1}])$ with $\boldsymbol{\alpha}'_{k+1} = \mathbf{s}_k^\alpha - \frac{1}{L_k} \mathbf{g}_{\boldsymbol{\alpha}}$ and $\boldsymbol{\beta}'_{k+1} = \mathbf{s}_k^\beta - \frac{1}{L_k} \mathbf{g}_{\boldsymbol{\beta}}$
6:   update $t_k = (1 + \sqrt{1 + 4t_{k-1}^2})/2$
7: **until** convergence (i.e., $|g(\boldsymbol{\alpha}_{k+1}, \boldsymbol{\beta}_{k+1}) - g(\boldsymbol{\alpha}_k, \boldsymbol{\beta}_k)| < \epsilon$)
8: **return** $\mathbf{w} = \frac{1}{\lambda \cdot m} (\boldsymbol{\alpha}_k^\top \mathbf{X}^+ - \boldsymbol{\beta}_k^\top \mathbf{X}^-)$

---

It should be noted that, (6) is a constrained problem, and therefore, at each step of gradient mapping, we have to project the dual solution into the domain $\Xi$ (i.e, $[\boldsymbol{\alpha}_{k+1}; \boldsymbol{\beta}_{k+1}] = \pi_\Xi([\boldsymbol{\alpha}'_{k+1}; \boldsymbol{\beta}'_{k+1}])$) in step 5) to keep them feasible. Below, we discuss how to solve this projection step efficiently.

**Projection Step** For clear notations, we expand the projection step into the problem

$$\min_{\boldsymbol{\alpha} \geq 0, \boldsymbol{\beta} \geq 0} \frac{1}{2} \|\boldsymbol{\alpha} - \boldsymbol{\alpha}^0\|^2 + \frac{1}{2} \|\boldsymbol{\beta} - \boldsymbol{\beta}^0\|^2 \quad \text{s.t. } \mathbf{1}_m^\top \boldsymbol{\alpha} = \mathbf{1}_n^\top \boldsymbol{\beta} , \tag{9}$$

where $\boldsymbol{\alpha}^0$ and $\boldsymbol{\beta}^0$ are the solutions obtained in the last iteration. We note that similar projection problems have been studied in [34, 24] where they either have $O((m+n)\log(m+n))$ time complexity [34] or only provide approximate solutions [24]. Instead, based on the following proposition, we provide a method which find the *exact* solution to (9) in $O(n+m)$ time. By using proof technique similar to that for Theorem 2 in [24], we can prove the following proposition:

**Proposition 1.** *The optimal solution to the projection problem in (9) is given by*

$$\boldsymbol{\alpha}^* = [\boldsymbol{\alpha}^0 - \gamma^*]_+ \quad and \quad \boldsymbol{\beta}^* = [\boldsymbol{\beta}^0 + \gamma^*]_+ ,$$

*where $\gamma^*$ is the root of function $\rho(\gamma) = \sum_{i=1}^m [\alpha_i^0 - \gamma]_+ - \sum_{j=1}^n [\beta_j^0 + \gamma]_+$ .*

Based on Proposition 1, we provide a method which find the exact solution to (9) in $O(m+n)$ time. According to Proposition 1, the key to solving this problem is to find the root of $\rho(\gamma)$. Instead of approximating the solution via bisection as in [24], we develop a divide-and-conquer method to find the exact solution of $\gamma^*$ in $O(m+n)$ time, where a similar approach has been used in [10]. The basic idea is to first identify the smallest interval that contains the root based on a modification of the randomized median finding algorithm [8], and then solve the root exactly based on the interval. The detailed projection procedure can be found in the longer version [22].

Table 1: Comparison of computational complexities for ranking algorithms, where $d$ is the number of dimensions, $\epsilon$ is the precision parameter, $m$ and $n$ are the number of positive and negative instances, respectively.

| Algorithm | | Computational Complexity |
|---|---|---|
| SVM$^{\text{Rank}}$ | [18] | $O\big(((m+n)d + (m+n)\log(m+n))/\epsilon\big)$ |
| SVM$^{\text{MAP}}$ | [42] | $O\big(((m+n)d + (m+n)\log(m+n))/\epsilon\big)$ |
| OWPC | [38] | $O\big(((m+n)d + (m+n)\log(m+n))/\epsilon\big)$ |
| SVM$^{\text{pAUC}}$ | [27, 28] | $O\big((n\log n + m\log m + (m+n)d)/\epsilon\big)$ |
| InfinitePush | [1] | $O\big((mnd + mn\log(mn))/\epsilon^2\big)$ |
| L1SVIP | [31] | $O\big((mnd + mn\log(mn))/\epsilon\big)$ |
| TopPush | this paper | $O\big((m+n)d/\sqrt{\epsilon}\big)$ |

## 3.3 Convergence and Computational Complexity

The theorem below states the convergence of the TopPush algorithm, which follows immediately from the convergence result for the Nesterov's method [29].

**Theorem 2.** *Let $\boldsymbol{\alpha}_T$ and $\boldsymbol{\beta}_T$ be the solution output from TopPush after $T$ iterations, we have*

$$g(\boldsymbol{\alpha}_T, \boldsymbol{\beta}_T) \leq \min_{(\boldsymbol{\alpha},\boldsymbol{\beta})\in\Xi} g(\boldsymbol{\alpha},\boldsymbol{\beta}) + \epsilon$$

*provided $T \geq O(1/\sqrt{\epsilon})$.*

Finally, since the computational cost of each iteration is dominated by the gradient evaluation and the projection step, the time complexity of each iteration is $O((m+n)d)$ since the complexity of projection step is $O(m+n)$ and the cost of computing the gradient is $O((m+n)d)$. Combining this result with Theorem 2, we have, to find an $\epsilon$-suboptimal solution, the total computational complexity of the TopPush algorithm is $O((m+n)d/\sqrt{\epsilon})$, which is linear in the number of training instances.

Table 1 compares the computational complexity of TopPush with that of the state-of-the-art algorithms. It is easy to see that TopPush is asymptotically more efficient than the state-of-the-art ranking algorithms[4]. For instances, it is much more efficient than InfinitePush and its sparse extension L1SVIP whose complexity depends on the number of positive-negative instance pairs; compared with SVM$^{\text{Rank}}$, SVM$^{\text{MAP}}$ and SVM$^{\text{pAUC}}$ that handle specific performance metrics via structural-SVM, the linear dependence on the number of training instances makes our TopPush approach more appealing, especially for large datasets.

## 3.4 Theoretical Guarantee

We develop theoretical guarantee for the ranking performance of TopPush. In [33, 1], the authors have developed margin-based generalization bounds for the loss function $\mathcal{L}_\infty^\ell$ . One limitation with the analysis in [33, 1] is that they try to bound the probability for a positive instance to be ranked before *any* negative instance, leading to relatively pessimistic bounds[5]. Our analysis avoids this pitfall by considering the probability of ranking a positive instance before *most* negative instances.

To this end, we first define $h_b(\mathbf{x}, \mathbf{w})$, the probability for any negative instance to be ranked above $\mathbf{x}$ using ranking function $f(\mathbf{x}) = \mathbf{w}^\top \mathbf{x}$, as

$$h_b(\mathbf{x}, \mathbf{w}) = \mathbb{E}_{\mathbf{x}^- \sim \mathcal{P}_-}\left[\mathbb{I}(\mathbf{w}^\top \mathbf{x} \leq \mathbf{w}^\top \mathbf{x}^-)\right] .$$

Since we are interested in whether positive instances are ranked above *most* negative instances, we will measure the quality of $f(\mathbf{x}) = \mathbf{w}^\top \mathbf{x}$ by the probability for any positive instance to be ranked below $\delta$ percent of negative instances, i.e.,

$$P_b(\mathbf{w}, \delta) = \Pr_{\mathbf{x}^+ \sim \mathcal{P}_+}\left(h_b(\mathbf{x}_i^+, \mathbf{w}) \geq \delta\right) .$$

Clearly, if a ranking function achieves a high ranking accuracy at the top, it should have a large percentage of positive instances with ranking scores higher than most of the negative instances, leading to a small value for $P_b(\mathbf{w}, \delta)$ with little $\delta$. The following theorem bounds $P_b(\mathbf{w}, \delta)$ for TopPush, and the detailed proof can be found in the longer version [22].

**Theorem 3.** *Given training data $S$ consisting of $m$ independent samples from $\mathcal{P}^+$ and $n$ independent samples from $\mathcal{P}^-$, let $\mathbf{w}^*$ be the optimal solution to the problem in (5). Assume $m \geq 12$ and $n \gg t$, we have, with a probability at least $1 - 2e^{-t}$,*

$$P_b(\mathbf{w}^*, \delta) \leq \mathcal{L}^\ell(\mathbf{w}^*, S) + O\big(\sqrt{(t + \log m)/m}\big)$$

*where $\delta = O(\sqrt{\log m/n})$ and $\mathcal{L}^\ell(\mathbf{w}^*, S) = \frac{1}{m}\sum_{i=1}^m \ell(\max_{1 \leq j \leq n} \mathbf{w}^\top \mathbf{x}_j^- - \mathbf{w}^\top \mathbf{x}_i^+)$.*

**Remark** Theorem 3 implies that if the empirical loss $\mathcal{L}^\ell(\mathbf{w}^*, S) \leq O(\log m/m)$, for most positive instance $\mathbf{x}_+$ (i.e., $1 - O(\log m/m)$), the percentage of negative instances ranked above $\mathbf{x}_+$ is upper bounded by $O(\sqrt{\log m/n})$. We observe that $m$ and $n$ play different roles in the bound; that is, because the empirical loss compares the positive instances to the negative instance with the largest score, it usually grows significantly slower with increasing $n$. For instance, the largest absolute value of Gaussian random samples grows in $\log n$. Thus, we believe that the main effect of increasing $n$ in our bound is to reduce $\delta$ (decrease at the rate of $1/\sqrt{n}$), especially when $n$ is large. Meanwhile, by increasing the number of positive instances $m$, we will reduce the bound for $P_b(\mathbf{w}, \delta)$, and consequently increase the chance of finding positive instances at the top.

## 4 Experiments

### 4.1 Settings

To evaluate the performance of the TopPush algorithm, we conduct a set of experiments on real-world datasets. Table 2 (left column) summarizes the datasets used in our experiments. Some of them were used in previous studies [1, 31, 3], and others are larger datasets from different domains. We compare TopPush with state-of-the-art algorithms that focus on accuracy at the top, including SVM$^{\text{MAP}}$ [42], SVM$^{\text{pAUC}}$ [28] with $\alpha = 0$ and $\beta = 1/n$, AATP [3] and InfinitePush [1]. In addition, for completeness, several state-of-the-art classification and ranking models are included in the comparison: logistic regression (LR) for binary classification, cost-sensitive SVM (cs-SVM) that addresses imbalance class distribution by introducing a different misclassification cost for each class, and SVM$^{\text{Rank}}$ [18] for AUC optimization. We implement TopPush and InfinitePush using MATLAB, implement AATP using CVX [14], and use LIBLINEAR [11] for LR and cs-SVM, and use the codes shared by the authors of the original works.

We measure the accuracy at the top by commonly used metrics[6]: (i) positives at the top (Pos@Top) [1, 31, 3], which is defined as the fraction of positive instances ranked above the top-ranked negative, (ii) average precision (AP) and (iii) normalized DCG scores (NDCG). On each dataset, experiments are run for thirty trials. In each trial, the dataset is randomly divided into two subsets: 2/3 for training and 1/3 for test. For all algorithms, we set the precision parameter $\epsilon$ to $10^{-4}$, choose other parameters by 5-fold cross validation (based on the average value of Pos@Top) on training set, and perform the evaluation on test set. Finally, averaged results over thirty trails are reported. All experiments are run on a machine with two Intel Xeon E7 CPUs and 16GB memory.

### 4.2 Results

In table 2, we report the performance of the algorithms in comparison, where the statistics of testbeds are included in the first column of the table. For better comparison between the performance of TopPush and baselines, pairwise $t$-tests at significance level of 0.9 are performed and results are marks "● / ○" in table 2 when TopPush is statistically significantly better/worse.

When an evaluation task can not be completed in two weeks, it will be stopped automatically, and no result will be reported. As a consequence, we observe that results for some algorithms are missing in Table 2 for certain datasets, especially for large ones. We can see from Table 2 that TopPush, LR and cs-SVM succeed to finish the evaluation on all datasets (even the largest datasets `url`). In contrast, SVM$^{\text{Rank}}$, SVM$^{\text{Rank}}$ and SVM$^{\text{pAUC}}$ fail to complete the training in time for several large datasets. InfinitePush and AATP have the worst scalability: they are only able to finish the smallest dataset `diabetes`. We thus conclude that overall, TopPush scales well to large datasets.

Table 2: Data statistics (left column) and experimental results. For each dataset, the number of positive and negative instances is below the data name as $m/n$, together with dimensionality $d$. For training time comparison,"▲" ("★") are marked if TopPush is at least 10 (100) times faster than the compared algorithm. For performance (mean±std) comparison, "•" ("○") is marked if TopPush performs significantly better (worse) than the baseline based on pairwise $t$-test at 0.9 significance level. On each dataset, if the evaluation of an algorithm can not be completed in two weeks, it will be stopped and its results will be missing from the table.

| Data | Algorithm | Time (s) | Pos@Top | AP | NDCG |
|---|---|---|---|---|---|
| diabetes<br>500/268<br>$d:34$ | **TopPush** | $5.11 \times 10^{-3}$ | $.123 \pm .056$ | $.872 \pm .023$ | $.976 \pm .005$ |
| | LR | $2.30 \times 10^{-2}$ | $.064 \pm .075\bullet$ | $.881 \pm .022$ | $.973 \pm .008$ |
| | cs-SVM | $7.70 \times 10^{-2}$ | $.077 \pm .088\bullet$ | $.758 \pm .166\bullet$ | $.920 \pm .078\bullet$ |
| | SVM$^{\text{Rank}}$ | $6.11 \times 10^{-2}$ | $.087 \pm .082\bullet$ | $.879 \pm .022$ | $.975 \pm .006$ |
| | SVM$^{\text{MAP}}$ | $4.71 \times 10^{0}$ | $.077 \pm .072\bullet$ | $.879 \pm .012$ | $.969 \pm .009$ |
| | SVM$^{\text{pAUC}}$ | $2.09 \times 10^{-1}$▲ | $.053 \pm .096\bullet$ | $.668 \pm .123\bullet$ | $.884 \pm .065\bullet$ |
| | InfinitePush | $2.63 \times 10^{1}$★ | $.119 \pm .051$ | $.877 \pm .035$ | $.978 \pm .007$ |
| | AATP | $2.72 \times 10^{3}$★ | $.127 \pm .061$ | $.881 \pm .035$ | $.979 \pm .010$ |
| news20-forsale<br>$999/18,929$<br>$d:62,061$ | **TopPush** | $2.16 \times 10^{0}$ | $.191 \pm .088$ | $.843 \pm .018$ | $.970 \pm .005$ |
| | LR | $4.14 \times 10^{0}$ | $.086 \pm .067\bullet$ | $.803 \pm .020\bullet$ | $.962 \pm .005$ |
| | cs-SVM | $1.89 \times 10^{0}$ | $.114 \pm .069\bullet$ | $.766 \pm .021\bullet$ | $.955 \pm .006\bullet$ |
| | SVM$^{\text{Rank}}$ | $2.96 \times 10^{2}$★ | $.149 \pm .056\bullet$ | $.850 \pm .016$ | $.972 \pm .003$ |
| | SVM$^{\text{MAP}}$ | $8.42 \times 10^{2}$★ | $.184 \pm .092$ | $.832 \pm .022$ | $.969 \pm .007$ |
| | SVM$^{\text{pAUC}}$ | $3.25 \times 10^{2}$★ | $.196 \pm .087$ | $.812 \pm .019\bullet$ | $.963 \pm .005\bullet$ |
| nslkdd<br>$71,463/77,054$<br>$d:121$ | **TopPush** | $7.64 \times 10^{1}$ | $.633 \pm .088$ | $.978 \pm .001$ | $.997 \pm .001$ |
| | LR | $3.63 \times 10^{1}$ | $.220 \pm .053\bullet$ | $.981 \pm .002$ | $.998 \pm .001$ |
| | cs-SVM | $1.86 \times 10^{0}$ | $.556 \pm .037\bullet$ | $.980 \pm .001$ | $.998 \pm .001$ |
| | SVM$^{\text{pAUC}}$ | $1.72 \times 10^{2}$ | $.634 \pm .059$ | $.956 \pm .002\bullet$ | $.996 \pm .001$ |
| real-sim<br>$22,238/50,071$<br>$d:20,958$ | **TopPush** | $1.34 \times 10^{1}$ | $.186 \pm .049$ | $.986 \pm .001$ | $.998 \pm .001$ |
| | LR | $7.67 \times 10^{0}$ | $.100 \pm .043\bullet$ | $.989 \pm .001$ | $.999 \pm .001$ |
| | cs-SVM | $4.84 \times 10^{0}$ | $.146 \pm .031\bullet$ | $.979 \pm .001$ | $.998 \pm .001$ |
| | SVM$^{\text{Rank}}$ | $1.83 \times 10^{3}$★ | $.090 \pm .045\bullet$ | $.986 \pm .000$ | $.999 \pm .001$ |
| spambase<br>$1,813/2,788$<br>$d:57$ | **TopPush** | $1.51 \times 10^{-1}$ | $.129 \pm .077$ | $.922 \pm .006$ | $.988 \pm .001$ |
| | LR | $3.11 \times 10^{-2}$ | $.071 \pm .053\bullet$ | $.920 \pm .010$ | $.987 \pm .003$ |
| | cs-SVM | $8.31 \times 10^{-2}$ | $.069 \pm .059\bullet$ | $.907 \pm .010\bullet$ | $.980 \pm .004\bullet$ |
| | SVM$^{\text{Rank}}$ | $2.31 \times 10^{1}$▲ | $.069 \pm .076\bullet$ | $.931 \pm .010$ | $.990 \pm .003$ |
| | SVM$^{\text{MAP}}$ | $1.92 \times 10^{2}$★ | $.097 \pm .069\bullet$ | $.935 \pm .014$ | $.984 \pm .005$ |
| | SVM$^{\text{pAUC}}$ | $1.73 \times 10^{0}$▲ | $.073 \pm .058\bullet$ | $.854 \pm .024\bullet$ | $.975 \pm .007\bullet$ |
| | InfinitePush | $1.78 \times 10^{3}$★ | $.132 \pm .087$ | $.920 \pm .005$ | $.987 \pm .002$ |
| url<br>$792,145/1,603,985$<br>$d:3,231,961$ | **TopPush** | $5.11 \times 10^{3}$ | $.474 \pm .046$ | $.986 \pm .001$ | $.999 \pm .001$ |
| | LR | $8.98 \times 10^{3}$ | $.362 \pm .113\bullet$ | $.993 \pm .001\circ$ | $.999 \pm .001$ |
| | cs-SVM | $3.78 \times 10^{3}$ | $.432 \pm .069\bullet$ | $.991 \pm .002$ | $.998 \pm .001$ |
| w8a<br>$1,933/62,767$<br>$d:300$ | **TopPush** | $7.35 \times 10^{0}$ | $.226 \pm .053$ | $.710 \pm .019$ | $.938 \pm .005$ |
| | LR | $2.46 \times 10^{0}$ | $.107 \pm .093\bullet$ | $.450 \pm .374\bullet$ | $.775 \pm .221\bullet$ |
| | cs-SVM | $3.87 \times 10^{0}$ | $.118 \pm .105\bullet$ | $.447 \pm .372\bullet$ | $.774 \pm .220\bullet$ |
| | SVM$^{\text{pAUC}}$ | $2.59 \times 10^{3}$★ | $.207 \pm .046$ | $.673 \pm .021\bullet$ | $.929 \pm .006\bullet$ |

**Performance Comparison** In terms of evaluation metric Pos@Top, we find that TopPush yields similar performance as InfinitePush and AATP, and performs significantly better than the other baselines including LR and cs-SVM, SVM$^{\text{Rank}}$, SVM$^{\text{Rank}}$ and SVM$^{\text{pAUC}}$. This is consistent with the design of TopPush that aims to maximize the accuracy at the top of the ranked list. Since the loss function optimized by InfinitePush and AATP are similar as that for TopPush, it is not surprising that they yield similar performance. The key advantage of using the proposed algorithm versus InfinitePush and AATP is that it is computationally more efficient and scales well to large datasets. In terms of AP and NDCG, we observe that TopPush yield similar, if not better, performance as the state-of-the-art methods, such as SVM$^{\text{MAP}}$ and SVM$^{\text{pAUC}}$, that are designed to optimize these metrics. Overall, we conclude that the proposed algorithm is effective in optimizing the ranking accuracy for the top ranked instances.

**Training Efficiency** To evaluate the computational efficiency, we set the parameters of different algorithms to be the values that are selected by cross-validation, and run these algorithms on full datasets that include both training and testing sets. Table 2 summarizes the training time of different algorithms. From the results, we can see that TopPush is faster than state-of-the-art ranking methods on most datasets. In fact, the training time of TopPush is similar to that of LR and cs-SVM

implemented by LIBLINEAR. Since the time complexity of learning a binary classification model is usually linear in the number of training instances, this result implicitly suggests a linear time complexity for the proposed algorithm.

**Scalability** We study how TopPush scales to different number of training examples by using the largest dataset `url`. Figure 1 shows the log-log plot for the training time of TopPush vs. the size of training data, where different lines correspond to different values of $\lambda$. For the purpose of comparison, we also include a black dash-dot line that tries to fit the training time by a linear function in the number of training instances (i.e., $\Theta(m + n)$). From the plot, we can see that for different regularization parameter $\lambda$, the training time of TopPush increases even slower than the number of training data. This is consistent with our theoretical analysis given in Section 3.3.

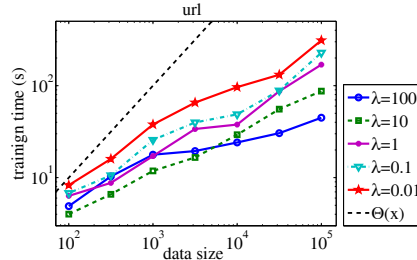

Figure 1: Training time of TopPush versus training data size for different values of $\lambda$.

## 5 Conclusion

In this paper, we focus on bipartite ranking algorithms that optimize accuracy at the top of the ranked list. To this end, we consider to maximize the number of positive instances that are ranked above any negative instances, and develop an efficient algorithm, named as TopPush to solve related optimization problem. Compared with existing work on this topic, the proposed TopPush algorithm scales linearly in the number of training instances, which is in contrast to most existing algorithms for bipartite ranking whose time complexities dependents on the number of positive-negative instance pairs. Moreover, our theoretical analysis clearly shows that it will lead to a ranking function that places many positive instances the top of the ranked list. Empirical studies verify the theoretical claims: the TopPush algorithm is effective in maximizing the accuracy at the top and is significantly more efficient than the state-of-the-art algorithms for bipartite ranking. In the future, we plan to develop appropriate univariate loss, instead of pairwise ranking loss, for efficient bipartite ranking that maximize accuracy at the top.

**Acknowledgement** This research was supported by the 973 Program (2014CB340501), NSFC (61333014), NSF (IIS-1251031), and ONR Award (N000141210431).

## Footnotes

[1]In this paper, we let $\ell(z)$ to be non-decreasing for the simplicity of formulating dual problem.

[2]Nonlinear function can be trained by kernel methods, and Nyström method and random Fourier features can transform the kernelized problem into a linear one. See [41] for more discussions.

[3] The step size of the Nesterov's method depends on the smoothness of the objective function. In current work we adopt the Nemirovski's line search scheme [29] to compute the smoothness parameter, and the detailed algorithm can be found in [22].

[4]In Table 1, we report the complexity of SVM$^{\text{pAUC}}_{\text{tight}}$ in [28], which is more efficient than SVM$^{\text{pAUC}}$ in [27]. In addition, SVM$^{\text{pAUC}}_{\text{tight}}$ is used in experiments and we do not distinguish between them in this paper.

[5]For instance, for the bounds in [33], the failure probability can be as large as 1 if the parameter $p$ is large.

[6] It is worth mentioning that we also measure the ranking performance by AUC, and the results can be found in [22]. In addition, more details of the experimental setting can be found there.

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
