[Reviews · NeurIPS 2014]

Submitted by Assigned_Reviewer_2

The authors present a novel approach to learning to rank. In contrast to traditional approaches, the idea is to focus on the number of positive instances that are ranked before the first negative one. Following a large-margin approach leads to primal and dual representations. Compared to similar approaches, the complexity is only linear in the number of instances.

This is a nice paper! Particularly the technical contribution is strong.

Apart from the url data set, the rest of the experiments is pretty small-scale. Adding experiments on larger-scales would certainly strengthen the paper as baseline competitors will at some point drop out. Consider having a figure showing performance vs number of training instances to showcase the benefit of processing more data than the baselines. The same figure for time instead of performance would also be interesting.

The divide and conquer schema should at least be sketched in the final paper.
Summary: Strong technical contribution. Good paper.

Submitted by Assigned_Reviewer_25

In this paper, the authors consider bipartite ranking and specifically, they consider optimizing the ranking at the top.
The authors first define a loss function that penalizes positive
example whenever its predicted score is less than that of the negative example with the highest
score. The main technical contribution is to propose a regularized optimization problem with
the above loss and to show that it can be solved efficiently via the dual formulation. The authors
give an optimization algorithm to solve the proposed formulation. They also give a theoretical result
that gives an upper bound on the probability that any positive example is ranked below delta fraction
of negative examples. The main advantage of the proposed formulation is that it brings down the computational
complexity of to rank optimization to linear in the number of examples.
Finally, the authors give experimental results comparing their approach with several other approaches. The proposed method clearly outperforms most other methods both in terms of speed and accuracy.

Minor comment:
The definitions of loss used in this paper is non-standard. Typically e^{-z} is the exponential
loss and [1-z]_{+}^2 is the truncated exponential loss. However, there is no issue since the
authors have also flipped the sign of the argument in Eqn (3). For clarity, I think that it is
better to modify this to match standard definitions.

The paper seems quite novel to me and the contributions in this paper seem non trivial. I do not
have any major concerns about this paper.
Summary: The authors propose an elegant approach to reduce the time complexity of bipartite ranking
to linear in the number of examples. The experimental results are quite compelling. I strongly recommend
accepting this paper.

Submitted by Assigned_Reviewer_31

The paper addresses the computational issue of bipartite ranking. The authors propose a new algorithm whose computational complexity is linear in number of training instances, and provide theoretical analysis of generalization error. The paper is rounded of with extensive experiments.

Strong points- The paper is clearly written and might be of some value in case of bipartite ranking with large datasets. The generalization bound is novel and experiments section is detailed.

However, i have multiple questions about this paper:
1. Though bipartite ranking is well studied, it is restricted in scope, in the sense that it is in the domain of ranking but cannot handle queries. Considering there are already well established algorithms for bipartite ranking which have been well studied theoretically and tested empirically, is the study really very valuable? For eg., this will only be useful when m,n are really large. Is that very practical in domain of bipartite ranking? Admittedly, this is just my thought and i would like to hear authors' view on this (citing example or something).
2. The main reason the paper gets an O(m+n/\sqrt(e)) error bound is because of the new defined target loss (2) and using a smooth convex surrogate which allows standard primal-dual trick to get a smooth dual, thereby allowing standard accelerated gradient descent optimization. According to target loss (2), a negative instance followed by all positive instances has greater loss than a positive instance followed by all negative instances. Can this be considered promoting good ranking at top, especially since there is no position based discount factor?

Moreover, in the supplement it is stated that maximizing positives at top cannot be achieved by AUC optimization, but target loss (2) is an upper bound on rank loss. So why should someone try to optimize an upper bound on rank loss, if AUC optimization itself is not suitable for the purpose of pushing positives at top?

3. Once the convex surrogate is taken to be smooth, conversion to dual and applying Nestrov technique is neat but i do not think it is extremely novel.

4. Looking at empirical section, i am confused as to what TopPush is gaining on LR (logistic regression). Computational power of the new algorithm is the USP of the paper; it does not seem to be doing any better than LR; infact LR takes less time than TopPush in 4/7 experiments. Nor is it gaining anything significant in AP and NDCG metrics, effectively metrics which are popular; it gains a little in position at top but loses in AUC. So why should we consider TopPush over LR? Or am i reading the experimental results wrong?

A side point: On comparing the computational complexity with SVM (Rank,MAP); it can be seen that all of them scale linearly with training data size. The gain in computation time in TopPush is because SVM consider hinge loss while TopPush considers a smooth surrogate. So computational complexity linear in number of training instances is not unique to TopPush.

5. Theory- In the generalization bound, shouldn't the focus be on the cases when there are a large number of negative instances and few positive instances? The other way round is less practical and even an average ranking function would put a few positive instances on the top. However, if we focus on the negative instances, i am not sure what the bound is relaying. With growing n, the empirical loss is much more likely to keep increasing, since the normalizing factor is only 1/m (no dependence on n). Since \delta will become small, the L.H.S probability is likely to grow but the R.H.S is also likely to grow. Maybe i am not being able to understand the significance of the bound, from a more useful n >> m point of view.

My ratings later on are provisional. I would like the authors to address the questions i have raised. Specifically i would like the authors to shed more light on the comparison between TopPush and LR ( 4, question on empirical section). In my opinion, the accept/reject hinges on clarifying how TopPush gains on LR. I will be happy to review my decision after author feedback.

Update after Author Feedback-
1. "Example of m,n large"- I am not an expert in bipartite ranking, so i will take the authors' words for it. However, from my knowledge of online advertisement, is it not the case that ranking online advertisements is in the learning to rank framework? (i.e query dependent?). I completely agree with the first reviewer that showing experiments for large datasets (possibly real datasets used in bi-partite ranking) will be very useful.
2. "AUC optimization"- The authors dont really answer the question. They talk about advantage of the new loss, in terms of optimization and gen. bounds. However, it has nothing to do with AUC. In fact, independent of how the new loss compares with AUC, the advantages will hold. From the point of view that the new loss is an upper bound on AUC and AUC is not useful for the objective of "pushing positives at top", why should someone optimize the new loss?
3."comparison with LR"- This is critical. I agree that TopPush is doing better than LR in Pos@Top metric. I think in the revised version, the authors should modify the introduction slightly. The USP of the paper is the computational advantage of TopPush over other algorithms. This is overselling the paper a little bit. TopPush has no (visible) computational advantage over LR. It can be seen as an alternate, with advantage when it comes to performance on a specific metric (and disadvantage on some others).
4."SVM"- I believe the advantage over SVM based methods is the quadratic convergence rate (O(1/T^2) as opposed to O(1/T)), not linear in "m+n"? Both SVM based methods and top push are linear in "m+n", as the authors have clearly shown in Table 1.
5."Gen bound"- Please include the discussion in the revised draft. This is critical.

Overall, i like this paper. With revision, it will certainly be a very good paper. I have updated my decision to an accept.

Summary: The problem addressed is well known with neat techniques used and might be of potential interest. However, there are some questions about the practical significance and the theoretical results. I do think it is an interesting paper and i will be happy to reconsider after authors' feedback; but right now, based on the nature of the highly competitive venue, i do not believe it will be a loss if NIPS gives it a miss.
Author Feedback
Author rebuttal: We want to thank reviewers and will improve the paper by incorporating the suggestions. In the following we will focus on technical questions.

Q1: examples where both m,n are large in practical bipartite ranking problems.

A1: Bipartite ranking has wide applications in information retrieval, recommender systems, etc. For example, in online advertisement system, we need to rank the advertisements that are more likely to be clicked at the top. In this case, both m and n (i.e. the number of clicked and non-clicked events) are very large.

Q2:“According to target loss (2) … Can this be considered promoting good ranking at top”

A2: Yes, in this paper, we focus on promoting ranking quality for the instances at the top of ranked list, which is different from AUC optimization. Specifically, we achieve this by the loss (2).

Q3:“why should someone try to optimize an upper bound on rank loss, if AUC optimization itself is not suitable for the purpose of pushing positives at top”

A3: The advantage of using loss in (2) is twofold. First, using loss function (2) will result in a dual problem with only $m+n$ dual variables, which is significantly smaller than $m*n$, the number of dual variables when using other loss function. Second, it will lead to a generalization error bound that emphasizes the error at the top of the ranking list.

Q4:“Once the convex surrogate is taken to be smooth, conversion to dual and applying Nestrov technique is neat but i do not think it is extremely novel.”

A4: There is an important technical trick we used to transform the dual problem into an optimization problem with only $m+n$ variables. We will make this clear in the revised draft.

Q5:“why should we consider TopPush over LR?” (emphasized by the 3rd reviewer)

A5: As shown in [Kotlowski et al. ICML (2011)], the logistic loss of LR is consistent to ranking loss, and as a result, LR can be seen as a method for optimizing AUC. In contrast, we emphasize optimizing the instances ranked at the top of a ranking list, which is beyond the optimization of AUC. This is demonstrated by our empirical study which shows that LR performs significantly worse than the proposed method if we measure the ranking results only based on the top ranked instances (i.e. Pos@Top).

Q6:“The gain in computation time in TopPush is because SVM(Rank, MAP) consider hinge loss while TopPush considers a smooth surrogate. So computational complexity linear in number of training instances is not unique to TopPush.”

A6: Hinge loss is critical to SVM(Rank, MAP) because it will result in a sparse dual solution, a key to make the cutting-plane method efficient. Replacing hinge loss in SVM(Rank, MAP) with a smooth loss will make it impossible to implement an efficient cutting plane method because the number of constraints is combinatorial in the number of training examples. Thus, complexity linear in the number of training examples is very unique to our approach.

Q7:“shouldn't the focus be on the cases when there are a large number of negative instances and few positive instances? …, if we focus on the negative instances, i am not sure what the bound is relaying. With growing n, the empirical loss is much more likely to keep increasing,…”

A7: We thank the reviewer for the insightful comment. Since the empirical loss only compares the positive instances to the negative instance with the largest score, it usually grows significantly slower with increasing $n$ (i.e. the number of negative instances). For instance, the largest absolute value of Gaussian random samples grows in $\log n$. Thus, we believe that the main effect of increasing $n$ in our bound is to reduce $\delta$ (decrease at the rate of $1/\sqrt{n}$), especially when $n$ is already very large.